# Can Cardiometabolic Risk Be Reduced in the Elderly? Comprehensive Epidemiological Study

**DOI:** 10.3390/geriatrics8040073

**Published:** 2023-07-01

**Authors:** Lavinia Maris, Timea Claudia Ghitea

**Affiliations:** 1Doctoral School of Biological and Biomedical Sciences, University of Oradea, 410087 Oradea, Romania; laviniamaris2010@yahoo.com; 2Faculty of Medicine and Pharmacy, Medicine Department, University of Oradea, 410068 Oradea, Romania

**Keywords:** cardiometabolic diseases, hypertension, physical activity, epidemiology

## Abstract

Through these epidemiological studies, which are based on statistical and observational calculations, without visual appeal, we tracked the incidence of public health problems. In this study, our research objective was to determine and evaluate the health patterns present in a population, along with identifying the factors that contribute to the risks or provide protection against specific diseases or conditions. The progression of cardiometabolic diseases is closely linked to various chronic conditions, such as diabetes, hypertension, dyslipidemia, and chronic kidney disease. This research study involved 578 patients, who were divided into six-year cohorts ranging from 2017 to 2022. The study examined parameters related to cardiometabolic diseases, including alcoholic hepatopathies, non-alcoholic hepatopathy, chronic kidney disease, hypertension, myocardial infarction, other forms of chronic coronary syndrome, peripheral vascular disease, microvascular diseases, macrovascular diseases, and hypercholesterolemia, while considering age and physical activity levels. The study concluded that individuals in the age group of 41–50 years exhibited the highest propensity for cardiometabolic damage. Additionally, the promotion of a healthy and active lifestyle is increasingly gaining traction among elderly patients.

## 1. Introduction

Although not visually appealing, epidemiological studies, which are based on statistical and observational analyses, are extremely valuable for monitoring large groups of individuals and identifying future trends. Regarding the study of cardiometabolic diseases, an epidemiological perspective, it highlighted their global importance as a public health problem [1]. Risk factors, including arterial hypertension (HTN), dyslipidemia, obesity, sedentary lifestyle, and unhealthy diet, exert a substantial influence on the development and progression of these diseases [2]. Comprehending and addressing these risk factors is crucial for the prevention and management of cardiometabolic diseases, as they hold significant potential for reducing the associated morbidity and mortality [3]. Therefore, epidemiological studies play a crucial role in identifying key aspects for each specific cohort.

The prevalence of cardiometabolic diseases [4,5] as revealed by epidemiological studies, has demonstrated a substantial increase in recent decades, particularly in middle- and high-income countries [5]. This rise can largely be attributed to lifestyle changes [6], unhealthy dietary habits [7], sedentary lifestyles, and obesity [8].

Significant risk factors for cardiometabolic diseases include HTN [9], dyslipidemia [10], smoking [11], diabetes, obesity, physical inactivity [12], and unhealthy diets [13]. Epidemiological studies have clearly established an association between these risk factors and the development of cardiometabolic diseases.

Cardiometabolic diseases have a considerable impact on public health by increasing both mortality and morbidity [14]. They contribute to cardiovascular complications such as myocardial infarction [15], stroke [16], and heart failure, as well as metabolic complications such as diabetic retinopathy and nephropathy [17].

Cardiovascular diseases, such as chronic coronary syndrome or congestive heart failure, as well as myocardial infarction, can be correlated with metabolic diseases [18,19]. In metabolic diseases such as type 2 diabetes or metabolic syndrome, heart disease often occurs as a comorbidity. The risk of these conditions increases with obesity, HTN, and visceral fat. A sedentary lifestyle also poses a risk factor. Dyslipidemia, particularly atherogenic dyslipidemia, increases the cardiovascular risk, exacerbates underlying metabolic diseases, and contributes to the development of new comorbidities [20,21].

Patients diagnosed with type 1 diabetes [22] and adolescent girls with polycystic ovary syndrome [23] have shown a heightened cardiometabolic risk, which has been associated with the presence of visceral obesity [24]. These metabolic conditions, directly associated with metabolic syndrome and type 2 diabetes, are closely linked to atherogenic dyslipidemia. The development of atherogenic dyslipidemia significantly diminishes both quality of life and life expectancy [25].

Cardiometabolic diseases have emerged as a global health issue, with a significant increase in prevalence and impact on the population in recent decades. Conditions such as coronary heart disease, stroke, diabetes, and metabolic syndrome demonstrate complex interconnections and mutual influences. The aim of this epidemiological study is to investigate the risk for atherosclerotic cardiovascular disease and evaluate their impact on public health. It also aims to determine whether the same approach is appropriate for both young and older people in reducing the risk of developing atherosclerosis. Starting from this premise, the ultimate goal is to develop a personalized approach to HTN, dyslipidemia, and metabolic diseases in young people, focusing on lifestyle modification and creating quality of life.

## 2. Materials and Methods

The study involved a total of 578 individuals with metabolic diseases, including metabolic syndrome and type 2 diabetes (DM2). They were divided into six groups based on the study year. The patients presented themselves at Arad County Emergency Clinical Hospital with hyperinsulinemia and hyperglycemia or diabetes, where cardiological, clinical, and paraclinical investigations were conducted. Each patient read the World Medical Association Declaration of Helsinki on Ethical Principles for Medical Research Involving Human Subjects [26]. Clinical and paraclinical analyses were performed to track the patients’ epidemiological progress, and the data were retrospectively processed. There were patients with both long-standing and new-onset hyperlipidemias. The clinical evaluation utilized the Tanita BC-545N body bioelectrical impedance analyzer available at the hospital to track fat mass and visceral fat. Changes in clinical and paraclinical parameters from the observation sheets were also monitored. The participants completed the evaluation forms at the beginning of the study period. Inclusion criteria required individuals to have at least three of the following medical conditions: obesity, visceral obesity, hypercholesterolemia, hypertriglyceridemia, HTN, myocardial infarction, congestive heart failure, chronic coronary syndrome, microvascular disease (microangiopathies showing chronic damage to small blood vessels), macrovascular disease (macroangiopathy means damage to large and medium arteries by the atherosclerosis process, including coronary artery disease, cerebrovascular disease and peripheral vascular disease), chronic kidney disease (any modification of the value outside the GFR values (glomerular filtration rate) 90 to 120 mL/min/1.73 m^2^), or alcoholic liver disease. Exclusion criteria included decompensated diabetes, renal stents, pacemakers, other cardiac or aortic medical devices, and autoimmune diseases. Due to the chronic nature of their conditions, patients were followed up for a minimum of 12 months (in mean 12.86 ± 0.12 months) with evaluations conducted every three months.

### 2.1. Statistical Analysis

Statistical analysis was performed using SPSS v20 and Microsoft Excel software packages. The initial step involved obtaining comprehensive descriptive statistics, which entailed calculating measures of central tendency and descriptive indicators and graphically presenting the most important findings. Using the Skewness and Kurtosis test (in the range of −3 to +3 for normally distributed data) we checked the normality of the data. A significance level of *p* < 0.05 was considered for all results. The Student’s *t*-test was utilized to calculate the mean values of parameters, frequency ranges, standard deviations, and perform statistical significance tests. The Spearman’s correlation coefficient was used to determine the independent indicator of units of measure for the two variables. A significance level of *p* < 0.05 indicated statistical significance for ANOVA, while a significance level of *p* < 0.01 denoted high-level statistical significance. Post hoc analysis (Bonferroni) was conducted as an additional subgroup analysis to analyze differences between groups.

### 2.2. Clinical Investigation

The clinical evaluation was conducted using the Tanita MC780MA body bio-electrical impedance analyzer from Germany, and the results were evaluated using medical software. BIA body analyzers are approved devices by the World Public Health Nutrition Association (WPHNA) that provide highly accurate measurements of body composition. The margin of error for these devices is 0.1 kg.

We monitored variations in four distinct groups based on sex, age, rural/urban environment, and clinical parameters such as BMI, visceral fat, hypercholesterolemia, hypertriglyceridemia, HTN, myocardial infarction, congestive heart failure, other forms of chronic coronary syndrome, peripheral vascular disease, macrovascular disease, microvascular disease, and chronic kidney disease, as well as alcoholic and non-alcoholic liver diseases, presented in the research flowchart (Figure 1).

## 3. Results

### 3.1. Demographic Description

A total of 578 patients were included in the research study, divided over six years. The average age of the participants ranged from 44.50 ± 12.37 to 50.71 ± 17.67, as shown in Table 1. The distribution of patients based on sex is as follows: 239 men (41.3%) and 339 women (58.7%). There were no significant differences observed in the distribution of patients by sex over the six-year period. The breakdown of patients by sex for each year is as follows: in 2017, there were 65 (14.1%) men and 96 (20.8%) women; in 2018, there were 31 (6.7%) men and 47 (10.2%) women; in 2019, there were 43 (9.3%) men and 66 (14.3%) women; and in the final year of the study (2020), there were 52 (11.3%) men and 62 (13.4%) women. Overall, the proportion of women in the cohort was 17.4% higher than that of men, with significant differences observed in each year of the study.

The patients were divided into six age categories to track epidemiological changes in the research parameters. This division is presented in Figure 2, categorized by gender. It can be observed that in the <30 years, 41–50 years, and 51–60 years age groups, there is a higher prevalence of women with metabolic problems. On the other hand, in the 31–40 years, 61–70 years, and >70 years age groups, men have a higher incidence.

Among the research parameters, the following findings were observed:

Elevated cholesterol levels were observed in 375 individuals, with the highest prevalence of hypercholesterolemia in 2020 (17.53%).

High triglyceride levels were recorded in 341 individuals in 2022, with the highest frequency at 20.27%.

HTN was present in 449 patients, with the lowest incidence in 2018 (15.21%).

Myocardial infarction was observed in 29 individuals, with the highest incidence of 24.82% in 2021.

Congestive heart failure was diagnosed in 147 out of 578 patients, with an incidence of 22.69% in 2019.

Other forms of chronic coronary syndrome were reported in a total of 239 individuals, with the highest incidence of 31.38% in 2021.

Peripheral vascular disease was identified in 164 patients, with the highest prevalence of 42.01% in 2021. Microvascular disease was present in 144 patients, while macrovascular disease was reported in only 74 patients.

Chronic kidney disease was observed in 361 individuals, accounting for 24.19% of the total, with the highest incidence in 2020.

Alcoholic liver diseases were present in 77.9% of patients, with the highest prevalence in 2022 (20.02%). Non-alcoholic liver diseases accounted for 46.7% of the total, with the highest incidence of 27.05% in 2022.

Statistical analysis using the Chi-square technique revealed significant differences (Table 2) within each age category as follows:

In individuals under the age of 30, non-alcoholic liver diseases (46.8%) and other forms of chronic coronary syndrome (32.3%) showed different trends over the 6-year study period.

Among individuals aged 31–40 years, there were no significant differences in myocardial infarction (9.4%) and the incidence of hypercholesterolemia (71.3%) over the 6-year study period. However, other parameters showed statistically significant differences (*p* < 0.05).

In the 41–50 age group, the incidence of non-alcoholic liver diseases (52.6%) differed significantly, and no significant differences in HTN (75.7%), myocardial infarction (1.3%), macrovascular disease (23%), and hypercholesterolemia (73%) were found over the 6-year study period.

Among individuals aged 51–60 years, significant differences were observed in the incidence of myocardial infarction (4.3%), and peripheral vascular disease (37.1%), and no significant differences in alcoholic liver diseases (75.7%) or in HTN (87.1%), over the 6-year study period.

In the 61–70 age group, there were no significant differences in the incidence of heart diseases, HTN (81.5%), myocardial infarction (6.2%), or alcoholic liver diseases (76.9%) across each year of the study.

For individuals over 70 years of age, liver diseases exhibited statistically significant variations depending on the year of study (*p* < 0.05), as did HTN (87.5%).

Congestive heart failure and triglyceride levels did not show statistically significant changes over the 6-year study period across the different age categories. Non-alcoholic liver disease and chronic coronary syndrome exhibited significant differences within each age category across all study years.

BMI reached its highest value in 2022, with a value of 31.32, indicating grade 1 obesity. The lowest BMI was observed in 2020, with a value of 29.38, which falls within the overweight category.

Visceral fat measurements showed an increase from 10.01 in 2020 to 11.54 in 2022.

Regarding physical activity, in 2021, the lowest percentage of individuals engaging in sports was 8.91% out of a total of 217 people. The highest percentage was recorded in 2022, with 24.85% of individuals participating in sports. The graphical representation of BMI, visceral fat, and physical activity in each age category based on the year of study is presented in Figure 3.

The study examined the progression of cardiometabolic diseases in relation to physical activity. The cohort was monitored, and it was observed that only 217 individuals (37.54%) engaged in some form of physical exercise, while 361 individuals remained sedentary. Sedentary lifestyle was considered a risk factor for cardiometabolic diseases and formed the basis of the evaluation. Table 3 presents the significant differences between physical activity and the research parameters.

The assessment of physical activity in the study was qualitative rather than quantitative. Therefore, any physical activity performed by the participants that exceeded a minimum of 30 min at least once a week was assigned a value of “1”. The absence of physical activity was denoted by a value of “0”.

From the graphical representation in Figure 4, the cardiometabolic parameters can be observed for each year of the study according to physical activity. It is evident that a decrease in physical activity was observed among patients with HTN in 2020 (Figure 4A), patients with myocardial infarction in 2020 (Figure 4B), 2021, and 2022, and among all patients with congestive heart failure (Figure 4C). Regarding patients with other forms of chronic coronary syndrome, an improvement in physical activity was noted in 2017, 2018, and 2019 (Figure 4D), while in other cases a decrease or complete lack of physical activity was observed. Additionally, Figure 4E shows a correlation between increasing BMI and physical activity, indicating that higher BMI values are associated with higher physical activity levels. However, a significant decrease in visceral fat was only observed in 2022 (Figure 4F).

Based on the non-parametric statistical evaluation using the Chi-square technique, a statistically significant difference was observed among patients under the age of 30 in terms of BMI (mean 29.78 ± 4.36). Furthermore, significant differences were found in patients aged between 31 and 40 years with congestive heart failure between the groups (26.5% of people) with physical activity (34.8% of people) and those without physical activity.

In patients aged over 60 (both between 61 and 70 years and over 70 years), with myocardial infarction (4 people between 61 and 70 years old, and 3 people over 70 years old) and congestive heart failure (16.9% between 61 and 70 years old, and 6.2% in people over 70 years old), significant differences were identified between the groups with physical activity (37.5%) and those without physical activity (62.5%). These findings are presented in detail in Table 4, which provides the Chi-square statistical description of the evolution of research parameters according to physical activity.

### 3.2. Correlations

Spearman’s correlation was used to analyze the relationships between all research parameters, presented in Table 5. A statistically significant, inversely proportional relationship was observed between congestive heart failure and age, as well as between age and visceral fat. This is evident from the negative value of the Spearman’s coefficient and the significance value of *p* < 0.05. The findings indicate that congestive heart failure tends to increase as age decreases, suggesting a higher risk for young patients. Similarly, a negative correlation was observed between age and visceral fat, indicating that higher levels of visceral fat were found in younger age groups. Additionally, in the current study, a positive correlation was found between physical activity and age, indicating that as individuals age, there is a greater emphasis on incorporating physical activity into a healthy lifestyle.

In the age group of 31–40, the incidence of congestive heart failure was found to be the highest. This observation can be attributed to the elevated levels of visceral fat, which is a significant cardiometabolic risk factor. Furthermore, individuals in this age group had a high BMI within the overweight category, with an average of 29.90 ± 4.26.

In the Cox regression analysis, a decrease of 1.7% in age was associated with a 6% increase in the risk rate of congestive heart failure (ExpB: 1.77, 95% CI: 1.27–2.47, *p* = 0.001). This increase was observed independently of several potential confounders, such as age, sex, hypertension (HTN), and dyslipidemias. The time to event parameter used in our test was one year. Survival analysis, analyzed by Cox regression, is a powerful analytical approach commonly employed in medical research to evaluate the risk of congestive heart failure over the course of the study period (Figure 5), presented in Table 6.

## 4. Discussion

Obesity is widely recognized as a significant health risk [27,28]. Numerous studies have explored the relationship between weight status and metabolic diseases and their therapeutical approaches [29,30]. Several studies have highlighted the direct link between obesity and metabolic syndrome [19,31,32], type 2 diabetes (T2DM) [8,33], with dysbiosis [34,35], and stress [36,37]. In our current study, we observed a higher prevalence of overweight and obesity among the patients. Metabolic syndrome involves an imbalance in carbohydrate metabolism [38], which is closely associated with increased visceral fat accumulation [39]. Previous research conducted in 2010 [24], highlighted the significant cardiovascular risk associated with increased visceral fat, while a study in 2018 [40] demonstrated a similar relationship with fat mass [41]. Since visceral fat is a major risk factor in metabolic diseases, this correlation helps explain the progression of cardiometabolic disorders. The assessment of visceral fat using a BIA analyzer has been compared with computed tomography (CT) assessment in several studies, and both methods have shown relevance in obtaining accurate results [42]. In our study, we observed a high level of visceral fat, reaching a value of 10, which is associated with an increased risk not only of cardiovascular diseases [10], but also of joint disorders [43], obstructive sleep apnea [44], and kidney diseases [45].

Atherogenic dyslipidemia has been extensively studied in Europe [5], China [46], and India. Our findings deviate significantly from the European studies, as we observed a dyslipidemia prevalence of 75% among the subjects from Arad, whereas Europe reported a prevalence of 55%. Hypercholesterolemia and hypertriglyceridemia, as components of dyslipidemia, are frequently evaluated together in most studies. In terms of gender, our study found a higher incidence of dyslipidemia in women compared to men, which is consistent with the findings reported in the literature. However, the observed differences did not reach statistical significance. Over the course of our study, we observed an increasing trend in the prevalence of dyslipidemia, particularly marked in hypercholesterolemia and hypertriglyceridemia. An epidemiological study from Malaysia [47], published in 2022, reported an incidence of 23.55% for hypercholesterolemia and 27.08% for hypertriglyceridemia, which are higher than our highest values of 17.53% for hypercholesterolemia and 20.27% for hypertriglyceridemia. This result can be explained by the fact that developing countries place greater emphasis on reducing health risks rather than treating complications, and this perspective is also reflected in the importance of conducting such studies in Romania in order to intervene and mitigate the risks.

Daily fat intake plays a crucial role in modifying the lipid profile [48]. Plasma saturated fat content is associated with dietary saturated fat, although endogenous sources can also contribute to its presence [49]. Dietary intake represents the primary source of Omega-6 and Omega-3 fatty acids [50].

Non-alcoholic liver diseases, which are detailed in numerous published papers [51,52], along with metabolic syndrome, affect approximately one third of the American population [53]. For patients over 60 years old, it is generally recommended to engage in adapted physical activity to reduce the risk of obesity, diabetes, and cardiovascular diseases. Lifestyle modifications, including exercise and diet, remain the mainstay of prevention [13,53,54,55], Therefore, it is crucial to closely monitor and recognize the imminent risk. Our study revealed a higher prevalence of non-alcoholic liver diseases, reaching 42% among the population in Arad.

In 2022, Talle conducted a study to track the prevalence of acute heart diseases [56]. The study recorded an incidence of 22.6% for acute myocardial infarction in Greece and 25% in Italy. These values are within the range of our data, with our recorded incidence of 24.82% being the highest in 2021. Additionally, the study also found a prevalence of 31.38% for chronic coronary syndrome.

The predisposition of the population to accumulate excess fat raises several concerns due to the increased incidence and prevalence of metabolic syndrome and T2DM. The alarming rise in the risk of developing peripheral vascular disease, whether microvascular or macrovascular, in individuals with these syndromes has necessitated closer monitoring. In addition to epidemiological assessments, we have also integrated interview directives [55]. The incidence of peripheral vascular disease increases with higher BMI, fat mass, and visceral fat accumulation [57]. Swenty’s study in 2020 focused on identifying the specific types of vascular diseases to establish the most effective strategies. A 2023 study tracked the prevalence of peripheral vascular disease among different age groups and found a significant increase with advancing age [58]. Over the course of the 4-year study, peripheral vascular disease was observed in 28% of the patients, indicating a significant rise in cardiovascular and circulatory risks. Furthermore, the incidence of peripheral vascular disease in 2021, at 42.01%, was higher than the average for the age group studied, suggesting a higher occurrence of the condition at younger ages. This finding highlights the importance of early detection and intervention in managing peripheral vascular disease and mitigating its associated risks.

Progressive renal damage has been observed in patients with dyslipidemia in a study conducted in 2017 [59]. Chronic kidney disease, a component of metabolic syndrome [60,61], is considered a consequence of the proinflammatory process [62]. It has also been correlated with coronary insufficiency [63], atherosclerosis [64], and cardiovascular mortality [65]. Some studies have identified chronic kidney disease as a risk factor primarily affecting men [66], while others have shown risks for both men and women [67]. In our study, we did not observe statistically significant differences between the sexes, although there were more women registered. Chronic kidney disease is strongly correlated with atherogenic dyslipidemia [68], and aligns with the pathophysiology of metabolic syndrome, posing a significant cardiovascular risk [69].

In 2022, Riazi published a study on non-alcoholic fatty liver disease (NAFLD) in 17 countries [70]. The study concluded a worldwide prevalence of 32.4% for NAFLD. In our own study, we observed a higher incidence of 27.05% in 2022, which was the highest recorded. The total prevalence in our study was 46.7%. This can be explained by the fact that the population included in our study may have already had existing risk factors for NAFLD, leading to a higher incidence compared to the global average. These findings emphasize the importance of understanding and addressing the risk factors associated with NAFLD to effectively manage and prevent the disease. 

Zheng et al. observed that higher levels of metabolic syndrome (MS) were associated with a hypertensive pattern [71]. In a study conducted in 2015, an increase in the prevalence of HTN and metabolic syndrome was observed, with a revealed correlation between the two conditions in regression analysis. Our study also noted a higher prevalence of obesity, as reflected by high BMI, among the patients, with 75% of them presenting HTN, 76% with alcoholic liver disease, and 62% with chronic kidney disease.

One limitation may stem from the observational nature of the study, where researchers solely observed and analyzed the data without intervening or controlling for variables. Another limitation could arise from confounding factors, referring to unaccounted factors that may alter the established correlations. Previous measurement or recording errors can also be considered as limitations since we lack access to verification or correction. Lastly, a significant limitation may pertain to external validity, meaning the study’s findings may not be applicable to diverse areas and regions. 

## 5. Conclusions

Based on the epidemiological study conducted on cardiometabolic risk, several conclusions can be drawn. Firstly, in 2022, the highest levels of visceral fat and BMI were recorded, indicating an increased risk of metabolic disorders. Additionally, there was a notable incidence of hypertriglyceridemia among the participants.

The practice of sports showed variations across different age categories. In 2022, older individuals engaged in sports activities to a greater extent. However, there was a significant discrepancy among age groups, particularly with the 41–50-years-old category showing the lowest participation in sports. This finding can be attributed to the high levels of stress experienced by individuals in this age group, leading to limited time for physical activity. Moreover, this age category exhibited a higher incidence of vascular and kidney diseases, further contributing to the cardiometabolic risk. It can be concluded that individuals between 41 and 50 years old face significant challenges in engaging in physical exercise, thereby increasing their susceptibility to cardiometabolic disorders and heightened mortality risk, especially considering the presence of oxidative stress. This study highlights the importance of promoting a healthy and active lifestyle, particularly among the elderly population, where a growing trend towards adopting such practices is observed.

## Figures and Tables

**Figure 1 geriatrics-08-00073-f001:**
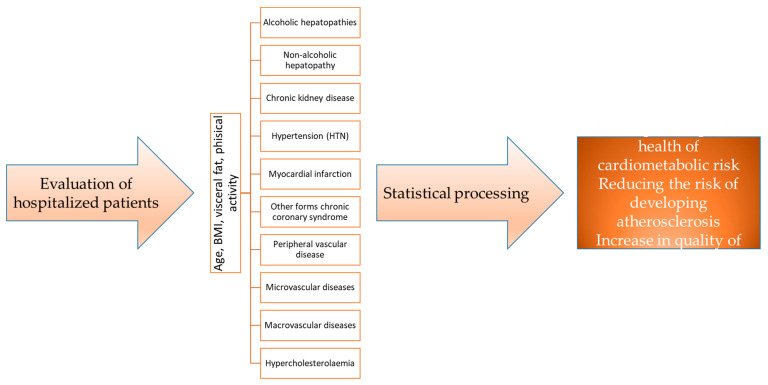
Research flowchart.

**Figure 2 geriatrics-08-00073-f002:**
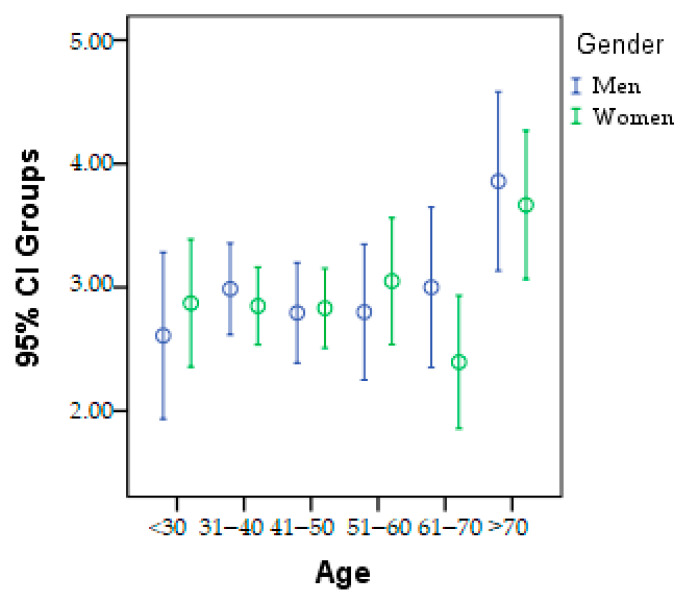
Graphic presentation of the age category according to gender.

**Figure 3 geriatrics-08-00073-f003:**
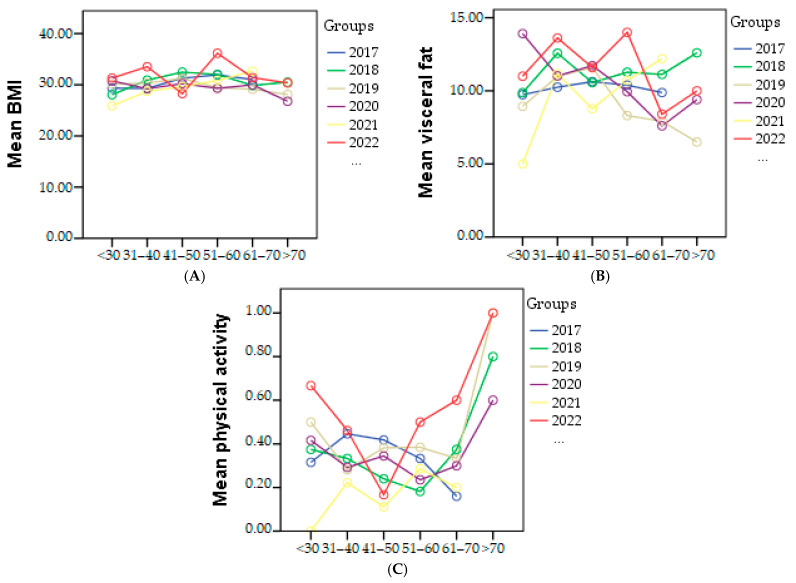
Graphical representation of mean of the BMI (**A**), visceral fat (**B**), and physical activity (**C**) in each age category based on the year of study.

**Figure 4 geriatrics-08-00073-f004:**
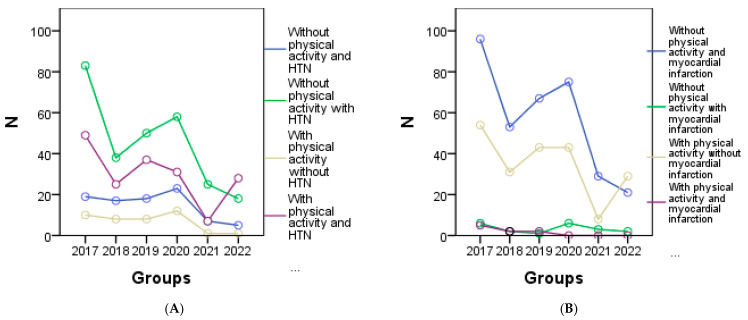
Graphical representation of the evolution of cardiometabolic parameters according to physical activity in each year of the study is as follows: (**A**) depicts individuals with or without HTN, and myocardial infarction (**B**), (**C**) shows individuals with congestive heart failure, and the (**D**) representing other forms of chronic coronary syndrome. (**E**) represents the evolution of BMI, and (**F**) the evolution of visceral fat for individuals with or without physical activity.

**Figure 5 geriatrics-08-00073-f005:**
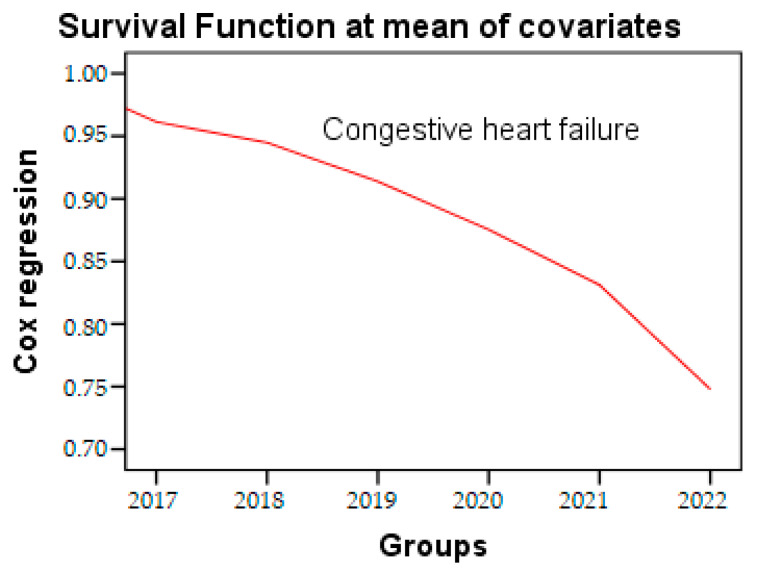
Cox regression of congestive heart failure during the research period.

**Table 1 geriatrics-08-00073-t001:** Demographic Description of the Cohort by Year of Study.

		N	%	Skewness		Kurtosis	
				Statistic	Std. Error	Statistic	Std. Error
Age category	<30	62	10.7	0.570	0.102	−0.589	0.203
	31–40	181	31.3				
	41–50	152	26.3				
	51–60	70	12.1				
	61–70	65	11.2				
	>70	48	8.3				
Age (mean/SD)	46.57 ± 14.15			0.641	0.102	−0.507	0.203
Gender	Men	239	41.3	−0.352	0.102	−1.882	0.203
	Women	339	58.7				
Origin	Urban	416	72.0	0.981	0.102	−1.041	0.203
	Rural	162	28.0				

N = number of patients, SD = standard deviation.

**Table 2 geriatrics-08-00073-t002:** Chi-square statistical analysis of the research parameters’ progression in each age group.

Parameters	Age											
	<30		31–40		41–50		51–60		61–70		>70	
	χ^2^	*p*	χ^2^	*p*	χ^2^	*p*	χ^2^	*p*	χ^2^	*p*	χ^2^	*p*
Alcoholic hepatopathies	6.419	0.268	13.315	0.021 *	6.705	0.244	2.700	0.746	4.364	0.498	8.436	0.038 *
Non-alcoholic hepatopathy	31.833	0.000 **	42.051	0.000 **	61.383	0.000 **	22.079	0.001 **	28.508	0.001 **	25.124	0.001 **
Chronic kidney disease	3.589	0.610	31.794	0.000 **	13.616	0.018 *	10.420	0.064	8.711	0.121	2.350	0.503
HTN	1.037	0.960	14.526	0.013 **	1.434	0.921	7.956	0.159	1.698	0.889	12.086	0.007 *
Myocardial infarction	0.000	1.000	4.925	0.425	2.826	0.727	11.656	0.040 *	4.214	0.519	6.893	0.075
Other forms of chronic coronary syndrome	17.472	0.004 **	41.969	0.000 **	49.176	0.000 **	21.268	0.001 **	17.433	0.004 **	12.384	0.006 *
Peripheral vascular disease	0.000	1.000	25.138	0.000 **	19.565	0.002 **	13.873	0.016 *	15.816	0.007 **	0.000	1.000
Microvascular diseases	0.000	1.000	25.521	0.000 **	26.712	0.000 **	10.498	0.062	15.816	0.007 **	0.000	1.000
Macrovascular diseases	0.000	1.000	15.056	0.010 *	7.328	0.197	5.475	0.361	3.362	0.644	0.000	1.000
Hypercholesterolemia	7.125	0.212	0.720	0.982	4.124	0.532	2.429	0.787	8.330	0.139	9.400	0.024 *

χ^2^ =coefficient Chi-square, *p* = statistically signification coefficient, ** Correlation is significant at the 0.01 level (2-tailed), * Correlation is significant at the 0.05 level (2-tailed).

**Table 3 geriatrics-08-00073-t003:** Progression of cardiometabolic parameters based on physical activity.

Parameters		Physical Activity			
		No		Yes	
		N	%	N	%
HTN	absent	89	24.65	40	18.43
	present	272	75.34	177	81.56
Myocardial infarction	absent	341	94.45	208	95.85
	present	20	5.54	9	4.14
Congestive heart failure	absent	258	71.46	173	79.72
	present	103	28.53	44	20.27
BMI (mean/sd)		29.99 ± 4.48		30.77 ± 5.28	
Visceral fat (mean/sd)		10.47 ± 4.14		10.95 ± 4.75	

N = number of patients.

**Table 4 geriatrics-08-00073-t004:** Chi-square statistical description of the evolution of research parameters according to physical activity.

Parameters	Age											
	<30		31–40		41–50		51–60		61–70		>70	
	χ^2^	*p*	χ^2^	*p*	χ^2^	*p*	χ^2^	*p*	χ^2^	*p*	χ^2^	*p*
Number of patients	578											
BMI	5.413	0.020 *	0.955	0.329	0.078	0.781	1.052	0.305	0.052	0.820	1.160	0.281
Visceral fat	0.001	0.971	3.338	0.068	0.153	0.696	0.581	0.446	0.040	0.841	4.300	0.038 *
HTN	0.008	0.928	1.648	0.199	0.110	0.740	1.730	0.188	2.712	0.100	1.549	0.213
Myocardial infarction	0.000	1.000	1.046	0.306	0.267	0.605	0.016	0.898	4.691	0.030 *	13.578	0.000 **
Congestive heart failure	0.029	0.865	5.590	0.018 *	0.023	0.881	0.359	0.549	4.695	0.030 *	13.578	0.000 **

χ^2^ =coefficient Chi-square, *p* = statistically signification coefficient, ** Correlation is significant at the 0.01 level (2-tailed), * Correlation is significant at the 0.05 level (2-tailed).

**Table 5 geriatrics-08-00073-t005:** The Spearman’s correlation results, including the research parameters and their relationship with age categories, along with the corresponding statistical significance.

Spearman’s Correlation		Age
Congestive heart failure	rho	−0.111 **
	*p*	0.008
Visceral fat	rho	−0.129 *
	*p*	0.002
Physical activity	rho	0.097 **
	*p*	0.007
N		578

N = number of patients, rho = coefficient Spearman’s, *p* = statistical significance, ** = Correlation is significant at the 0.01 level (2-tailed), * = Correlation is significant at the 0.05 level (2-tailed).

**Table 6 geriatrics-08-00073-t006:** Cox regression of parameters during the research period.

Variables in the Equation							
Parameters	B	SE	Wald	Sig.	Exp(B)	95.0% CI for Exp(B)	
						Lower	Upper
Congestive heart failure	0.573	0.170	11.369	0.001	1.774	1.271	2.475
Visceral fat	−0.087	0.033	6.811	0.009	0.917	0.859	0.979
Physical activity	−0.582	0.280	4.315	0.038	0.559	0.322	0.968

## Data Availability

All the data processed in this article are part of the research for a doctoral thesis, being archived in the aesthetic medical office, where the interventions were performed.

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
