# Peer review of "Can Cardiometabolic Risk Be Reduced in the Elderly? Comprehensive Epidemiological Study"

_geriatrics, 2023, doi:10.3390/geriatrics8040073_

Round 1
Reviewer 1 Report
This is interesting research concerning the issue of cardiometabolic diseases which is an ongoing epidemic with multifaceted health effects undermining individual and social well-being. Generally, this is a concise, and well-written manuscript. It is in line with the aim and objectives of the journal. The language of the paper is cohesive, free from jargons using appropriate academic language, clarity, and grace. The literature is relevant to the topic. The authors have collected a very good data. The results are clear and well-written. Tables and figures show properly the data in details. The paper conveys information concisely avoiding overload. The authors make a strong contribution to the research literature in this specific area of investigation. Overall, this is a good quality manuscript that can offer important information to support further research and experimentation for the theoretical and practical basis of cardiometabolic diseases and his risk factors.
Areas of weakness.
Abstract:
The text between lines 9 – 13 is not adequate for the objectives and results of the research carried out.
It does not indicate the objectives of the research, nor the most relevant results.
Introduction:
Since the objective of the work is not to demonstrate the effectiveness/efficiency of epidemiological studies in the investigation of risk factors associated with metabolic syndrome and the control of its risk factors, in the introduction part the text between lines 25 – 28 is not adequate for the objectives and results of the research carried out.
The hypertension term appears for the first time in line 30, it should be specified that it is arterial hypertension and indicate its acronym at this time and be used in future.
Reference 4 (line 37) could be merged with reference 5 (line 39) since the first statement is not relevant to being referenced.
The text between lines 56-59 must be corrected, as well as its bibliographical references.
Aims:
From line 65 to 72 they are mentioned several times, not coinciding in their wording, and therefore it is not clear what the purposes of the study are.
Materials and Methods:
The variables that are measured or how they are collected are not indicated. There is also no research flowchart included.
It is not indicated whether the participants were informed of the objectives of the study or compliance with ethical standards in the research.
Results
The text of lines 233-234 seems more suitable for discussion than for indicating results. The same thing happens with lines 255-257.
Discussion
Ιn the discussion it is recommended adding more information regarding research innovation, significant results, practical implications, and suggestions for future research. More specific, it is suggested that more should be written about the general contribution of this. It is recommended to rearrange the paragraphs and rewrite them to improve their association with the objectives and their understanding.
References
25% of the cited references are older than 10 years, it is recommended to reduce this percentage.
Author Response
Response to Reviewer 1
Firstly, I, the author of the present manuscript wish to thank you for thoughtful commentary you have provided to improve the quality of the paper. I am very grateful for the time and effort you have devoted to this task. We have extensively revised my manuscript according to the recommendations. All changes in the text and the new figures that we have redesigned are highlighted. Please, see the point-by-point answers to your comments below. All correction was highlighted in the manuscript.
Comments:
This is interesting research concerning the issue of cardiometabolic diseases which is an ongoing epidemic with multifaceted health effects undermining individual and social well-being. Generally, this is a concise, and well-written manuscript. It is in line with the aim and objectives of the journal. The language of the paper is cohesive, free from jargons using appropriate academic language, clarity, and grace. The literature is relevant to the topic. The authors have collected a very good data. The results are clear and well-written. Tables and figures show properly the data in details. The paper conveys information concisely avoiding overload. The authors make a strong contribution to the research literature in this specific area of investigation. Overall, this is a good quality manuscript that can offer important information to support further research and experimentation for the theoretical and practical basis of cardiometabolic diseases and his risk factors.
Areas of weakness.
Abstract:
Comment 1. The text between lines 9 – 13 is not adequate for the objectives and results of the research carried out. It does not indicate the objectives of the research, nor the most relevant results.
Answer 1. Thank you very much for the comment. I completed the manuscript (lines 9-13)
Introduction:
Comment 2. Since the objective of the work is not to demonstrate the effectiveness/efficiency of epidemiological studies in the investigation of risk factors associated with metabolic syndrome and the control of its risk factors, in the introduction part the text between lines 25 – 28 is not adequate for the objectives and results of the research carried out.
Answer 2. Thank you very much for the suggestion, the obiectives were modified accordingly (lines 26-30).
Comment 3. The hypertension term appears for the first time in line 30, it should be specified that it is arterial hypertension and indicate its acronym at this time and be used in future.
Answer 3. Again, we agree with your suggestion, and correction was made accordingly in all manuscript.
Comment 4. Reference 4 (line 37) could be merged with reference 5 (line 39) since the first statement is not relevant to being referenced.
Answer 4. Thank you very much for the suggestion, the references was merged (lines 37, 39).
Comment 5. The text between lines 56-59 must be corrected, as well as its bibliographical references.
Answer 5. Thank you for the amendment. I complete the manuscript with the required information. Please, see the correction highlighted in the manuscript (lines 56-58).
Aims:
Comment 6. From line 65 to 72 they are mentioned several times, not coinciding in their wording, and therefore it is not clear what the purposes of the study are.
Answer 6. Thank you for the suggestion. I corrected the manuscript (lines 65-71). Please, see the correction highlighted in the manuscript.
Materials and Methods:
Comment 7. The variables that are measured or how they are collected are not indicated. There is also no research flowchart included.
Answer 7. Thank you for the amendment. I complete the manuscript with the required information and with research flowchart. Please, see the correction highlighted in the manuscript (lines 75-76, 121-122).
Comment 8. It is not indicated whether the participants were informed of the objectives of the study or compliance with ethical standards in the research.
Answer 8. Thank you for the remark. A great emphasis was placed on informing the patient, the agreement regarding the processing of personal data was signed, and they were informed about the objectives of the study as well as the compliance with ethical standards in research from Helsinki. The manuscript was completed with the required information. Lines 77-79.
Results
Comment 9. The text of lines 233-234 seems more suitable for discussion than for indicating results. The same thing happens with lines 255-257.
Answer 9. Thank you for the amendment. The sentences was moved in discussion (lines 296-298, 325-3270
Discussion
Comment 10. In the discussion it is recommended adding more information regarding research innovation, significant results, practical implications, and suggestions for future research. More specific, it is suggested that more should be written about the general contribution of this. It is recommended to rearrange the paragraphs and rewrite them to improve their association with the objectives and their understanding.
Answer 10. Thank you very much for the amendment. The discussin was rearranged and completed.
References
Comment 11. 25% of the cited references are older than 10 years, it is recommended to reduce this percentage.
Answer 11. Thank you for the remark. The references were completed with newest references, and the older were replaced.

Author Response
Response to Reviewer 2
Firstly, I, the author of the present manuscript wish to thank you for thoughtful commentary you have provided to improve the quality of the paper. I am very grateful for the time and effort you have devoted to this task. We have extensively revised my manuscript according to the recommendations. All changes in the text and the new figures that we have redesigned are highlighted. Please, see the point-by-point answers to your comments below. All correction was highlighted in the manuscript.
Comments:
The study aimed to investigate a section of the population of Arad regarding cardiometabolic disorders and age-groups. The researchers collected data about 578 patients with hyperglycemia or hyperinsulinemia during a 6-year period in every 3 months. The investigational period was between 2017-2022. To continue the study even during COVID pandemic, makes the results very precious. The study is very valuable, offers a clear chance for epidemiological analyses, however most results are presented in a highly confusing manner.
Introduction:
Comment 1. In line 56-57 there is a non-English part in the text. Please revise!
Answer 1. Thank you for the amendment. I corrected the manuscript. Please, see the correction highlighted in the manuscript (lines 56-58).
Materials and methods:
Comment 2. The investigation is obviously fit for a population analyses, but it is recommended to mention e.g. in a „Limitations” section, that it is „only” a single-center, observational study with some selection bias. The main inclusion criteria were hyperglycemia or hyperinsulinemia at the first presentation and 3 of the following conditions: obesity, visceral obesity, hypercholesterolemia, hypertriglyceridemia, hypertension, myocardial infarction, congestive heart failure, ischemic heart disease, microvascular disease, macrovascular disease, chronic kidney disease, or alcoholic liver disease. What about the main cardiometabolic disease, diabetes?
Answer 2. Thank you for the remark. The study was based on a larger study that included patients with cardiometabolic diseases, of course also diabetes. The manuscript was completed with this detail (line 76).
Comment 3. The inclusion criteria need to be refined.
- Hypercholesterolemia presumably refers to total cholesterol level. There are very important treatment goals for diabetic, cardiovascular and vascular patients. Did you use these guidelines recommended levels of cholesterol (e.g. <200 mg/dL in cardiovascular patients) to diagnose high cholesterol or your laboratory values for high cholesterol level, which is given for a certain but definitely healthy population? I have to make the same question regarding high triglyceride levels as well. Although the manuscript did not give any informations about antilipid medication, it could be relevant, as patients taking antilipid medication have a reduced risk for cardiovascular events but have a higher cardiovascular and cardiometabolic risk then healthy ones. Do the investigators have any informations about the antilipid medication of their patients?
Answer 3. Thank you for the remark. The manuscript was completed with the detailed description of the research groups, and with the inclusion / exclusion criteria. The treatment was not specified because the patients who arrived at the emergency hospital in Arad either were not receiving treatment (due to non-adherence or being newly diagnosed with cardiovascular problems) or had been on the same treatment for an extended period. However, the new treatment recommended by the hospital aligns with guideline recommendations for managing hypercholesterolemia and hypertriglyceridemia. We, the authors, completed the maanuscript with this specification (lines 96-98).
Comment 4. The manuscript mentioned several times micro, - and macrovascular diseases, but do not define it. Please specify what are these terms referring to. Whether it should be a comprehensive term for all vascular disease it could give false positive results for your inclusion, where you have all the main cardiovascular diseases listed again.
Answer 4. Thank you for the amendment. I complete the manuscript with the required information. Please, see the correction highlighted in the manuscript (lines 87-91).
„microvascular disease (microangiopathies showing chronic damage to small blood vessels), macrovascular disease (macroangiopathy means damage to large and medium arteries by the atherosclerosis process, including coronary artery disease, cerebrovascular disease and peripheral vascular disease),”
Comment 5. Among exclusion criteria “stents” are listed. Is this word referring to aortic or renal stents or even to coronary interventions? The question is pretty important as among inclusion criteria the paper lists myocardial infarction, and according to the recent cardiology guidelines the primary care for acute coronary syndrome is coronary catheter intervention and stent implementation. Did you involve only patients with myocardial infarction treated conservatively? Please specify!
Answer 5. Thank you very much for the amendment. I complete the manuscript with „renal” stents (line 93).
Comment 6. Based on the 2019. ESC and AHA/ACC guideline it is recommended to use the phrase chronic coronary syndrome instead of ischemic heart disease, except there are some special reasons to use this idiom. Please explain or replace in the manuscript!
Answer 6. Gratefully accepting the observation, we, the authors, replaced ischemic heart disease in chronic coronary syndrome in all manuscript.
Comment 7. Please define, what chronic kidney disease is referring to! Did you choose a specific GFR value?
Answer 7. Thank you very much for observation. In monitoring the health of the entire population, not just a specific section of the emergency hospital in Arad, chronic kidney disease is defined as having a glomerular filtration rate (GFR) below 90 mL/min/1.73 m², rather than within the range of 90 to 120 mL/min/1.73 m². This clarification completes the information provided in the manuscript, specifically in lines 91-92.
Comment 8. There is no information about the informed consent and the ethical license. Please provide!
Answer 8. The study was conducted in accordance with the Declaration of Helsinki and approved by the Institutional Review Board (or Ethics Committee) of the University of Oradea (protocol code CEFMF/1 from 31 January 2023 and date of approval). Lines 405-407
Comment 9. It is mentioned, that all the patients participated in the study minimum 12 months long, but it would be more informative to describe the average of the follow-up period for your investigated population. Please specify!
Answer 9. Again, we agree with your suggestion, and correction was made accordingly (line 95).
Comment 10. You don not answer this question “At the conclusion of the study, a risk analysis was conducted to determine if the absence of oral antidiabetic treatment could be considered a risk factor. For this purpose, the odds ratio (OR) parameter, 95% confidence interval (CI), and Chi-square test were employed”. Please remove or provide and discuss the results.
Answer 10. Thank you very much for the amendment. I removed this sentence from the manuscript.
Comment 11. Please describe the normality check test you have choose for your continuous data.
Answer 11. Thank you for observation. I completed the manuscript with the required informations. (lines 104-105)
„Using the Skewness and Kurtosis test (in the range of -3 to +3 for normally distributed data) we checked the normality of the data.”
Results:
Comment 12. The study offers plenty of important results, but the number of the included patients and the gender rates in certain years is not one of them. Please remove this part from demographic description as well as from Table 1, and give information about the research parameters (age, gender, origin etc) during the whole investigational period. You are referring very often later to the number of patients in the certain age groups, but you never give these certain numbers. Please provide information! How much patients did you recruit in the age groups?
Answer 12. Thank you for remark! The table 1 has been revised to include the requested data, including the number of patients in each age group. Please, see the correction highlighted in the manuscript in line 140.
Comment 13. The investigators described in Results section very detailed in which year the research parameters reached the highest value or the highest incidence, but do not discuss these results. If they are not worth for discussion, remove please, if they are, please, discuss the significance.
Answer 13. Thank you very much for the amendment. I completed the discussion. (lines 316-322, 334-338, 346-354, 365-371)
Comment 14. From line 164 please specify the incidence and prevalence values for each research parameters, as readers could only draw the fact from chi square test results, that there was a significant relation between the age and the certain parameter.
Answer 14. Thank you for the amendment. I complete the manuscript with the required information. Please, see the correction highlighted in the manuscript (lines 173-192).
Comment 15. The description of the results from correlation analyzes is quite confusing. According to Table 5 you have investigated the relation of age to the research parameters, but the description suggests, that you have analyzed the relation of congestive heart failure and visceral fat as well (which could be very interesting, by the way). However, there is a way to analyze the association of categorical or dichotomous variables by correlation (e.g. point-biserial correlation), but Pearson correlation is not appropriate. So if you want to analyze your continuous and categorical variables with correlation please choose an appropriate method.
Answer 15. Thank you very much for the amendment. I changed test statistic in the manuscript with the Spearmen`s correlation. (lines 259-279)
Comment 16. The mentioned (and presumably inappropriately used) Pearson correlation indicates only a link between the investigated parameters, but to validate, that age, physical activity or visceral fat should be independent determining factors for congestive heart failure you need to analyze the data by regression. Correlation is definitely not enough to draw your conclusion about age, physical activity and congestive heart failure. As you have all the valuable data to perform a multiple linear or even a Cox regression please perform.
Answer 16. Thank you very much for the amendment. I complete the manuscript with the Cox regression. (lines 280-286)
Comment 17. How would you explain, that based on your data congestive heart failure is more often in the younger population?
Answer 17. Thank you very much for observation! In the age group of 31-40, the incidence of congestive heart failure was found to be the highest. This observation can be attributed to the elevated levels of visceral fat, which is a significant cardiometabolic risk factor. Furthermore, individuals in this age group had a high BMI within the overweight category, with an average of 29.90±4.26. (Lines 355-358)
Comment 18. The heading for Table 2 is not correct. It is not referring to progression of the research parameters in each year, but in each age group. Please correct or redraw the table.
Answer 18. Thank you for the amendment. I corrected the heading of table 2 (line 198)
Comment 19. The results about physical activity are very important, as they show, that the population mostly affected by cardiovascular diseases, are performing the less physical activity. How did you calculate mean physical activity?
Answer 19. Thank you for the amendment. I complete the manuscript with the required information. Please, see the correction highlighted in the manuscript (lines 222-226).
The assessment of physical activity in the study was qualitative rather than quantitative. Therefore, any physical activity performed by the participants that exceeded a minimum of 30 minutes at least once a week was assigned a value of "1". The absence of physical activity was denoted by a value of "0".
Comment 20. Line 278: missing “I”.
Answer 20. Thank you very much for the amendment. I corrected them. (line 303)
Comment 21. Line 316: missing period after [56].
Answer 21. Thank you very much for the amendment. I corrected them. (line 364)
Comment 22. Please discuss your results in more detail and related to your results and conclusions. Why is hypertriglyceridemia, hypercholesterolemia, non-alcoholic liver disease more prevalent in Arad? Based on your data, is there a link between more dyslipidemia and more new-onset of diabetes or cardiovascular conditions?
Answer 22. Thank you very much for comment! The epidemiological study was conducted by the authors, who are doctors, and they have observed an increase in the incidence of diabetes and cardiometabolic diseases in recent years. Several studies have been conducted to investigate the underlying causes of these increases, focusing on lifestyle and dietary factors, including the impact of excessive intake of ultra-processed foods. Each study provides crucial information to gather meaningful results in order to address not only your question but also our inherent curiosity. We hope to outline the answer in the near future.
In summary, the manuscript is valuable, but needs extensive refinements is statistical analyzes, presentation and discussion. As a consequence of the above mentioned questions and comments my suggestion is major revision.
Hopefully you will succeed in publishing your work.
With best regards.

Round 2
Reviewer 1 Report
This is interesting research concerning the issue of cardiometabolic diseases which is an ongoing epidemic with multifaceted health effects undermining individual and social well-being. Generally, this is a concise, and well-written manuscript. It is in line with the aim and objectives of the journal. The language of the paper is cohesive, free from jargons using appropriate academic language, clarity, and grace. The literature is relevant to the topic. The authors have collected a very good data. The results are clear and well-written. Tables and figures show properly the data in details. The paper conveys information concisely avoiding overload. The authors make a strong contribution to the research literature in this specific area of investigation. Overall, this is a good quality manuscript that IT offer important information to support further research and experimentation for the theoretical and practical basis of cardiometabolic diseases and his risk factors.
The authors of the present manuscript have extensively revised my suggestions.
Author Response
The author of the present manuscript wish to thank you for thoughtful commentary you have provided to improve the quality of the paper. I am very grateful for the time and effort you have devoted to this task.
Thank you for your positive response.
Reviewer 2 Report
2. Review for geriatrics-2446968
Dear authors! Thank you very much for your excellent, proper and fast reactions for my review. I really appreciate for taking into account my suggestions, and performing the new statistical analyzes too. The new research map is very helpful for understanding your work flow.
Line 68: would be better: the risk for atherosclerotic cardiovascular disease. You did not do research on atherosclerosis, what is there in your patients, it will not develop anymore.
Line 77: Your patient signed the informed consent not the Declaration of Helsinki.
Table 1: at age should be a tip failure: 4,57 could not be average age.
Materials and methods:
I could not find the Limitation part, or the fact that your study has a single-center manner.
I still cannot find the lipid targets for your inclusion criteria, or any definition for this. I understand, that you used the guideline mentioned goals for the diagnosis and treatment, but you mentioned in your answer, that there were patients with long-standing and patients with new-onset of hyperlipidemias. At least, this fact should be mentioned in the manuscript, whether you cannot afford a value.
Results:
In line 105 is still Pearson correlation. Please correct it! The Spearman’s correlation is a better tool for analyze your data, but still the presentation in the text and the table you have inserted is not about the same. Your Table 5 shows data about the association between age and the parameters (congestive heart failure, visceral fat, physical activity), and not about congestive heart failure and visceral fat. Anyway, the analyzes about visceral fat and age, Pearson was a good choice, as these parameters are measured at the continuous scale.
Thank you for rewrite the chi square test results, but it still not suitable. You now describe in the text and also in the table the same information, which is not necessary. Please provide in the Table 2 and 4 the number (N) of patients in each groups, or at least the percentage. Like you did it in case of Table 3.
You have a binary scale for describing physical activity. In this case I don’t understand the phrase “mean” physical activity. I think you wanted to describe the average number of patients who are participating in physical activity and not the average of a nominal variable. Please correct the graph, maybe it would be better to add some units to avoid misunderstandings.
For the Cox regression: Thank you for taking time to perform this analyzes. What is the time-to event parameter in your test? If it is new onset of congestive heart failure, it could not be also a variable in the equation. Please revise the description and your results!
Sincerely Yours,
Author Response
Response for reviewer 2, round 2
Firstly, I, the author of the present manuscript wish to thank you for thoughtful commentary you have provided to improve the quality of the paper. I am very grateful for the time and effort you have devoted to this task. We have extensively revised my manuscript according to the recommendations. All changes in the text and the new figures that we have redesigned are highlighted in yellow. Please, see the point-by-point answers to your comments below. All correction was highlighted in the manuscript.
- Review for geriatrics-2446968
Dear authors! Thank you very much for your excellent, proper and fast reactions for my review. I really appreciate for taking into account my suggestions, and performing the new statistical analyzes too. The new research map is very helpful for understanding your work flow.
Comment 1. Line 68: would be better: the risk for atherosclerotic cardiovascular disease. You did not do research on atherosclerosis, what is there in your patients, it will not develop anymore.
Answer 1. Thank you for the remark. The manuscript was corrected. (line 68)
Comment 2. Line 77: Your patient signed the informed consent not the Declaration of Helsinki.
Answer 2. Thank you for the amendment. We, the authors, have removed „and signed”. (line 77)
Comment 3. Table 1: at age should be a tip failure: 4,57 could not be average age.
Answer 3. Thank you for the observation. The manuscript was corrected.
Materials and methods:
Comment 4. I could not find the Limitation part, or the fact that your study has a single-center manner.
Answer 4. Thank you for the remark. The manuscript was completed with the limitations. (lines 378-384)
„One limitation may stem from the observational nature of the study, where researchers solely observed and analyzed the data without intervening or controlling for variables. Another limitation could arise from confounding factors, referring to unaccounted factors that may alter the established correlations. Previous measurement or recording errors can also be considered as limitations since we lack access to verification or correction. Lastly, a significant limitation may pertain to external validity, meaning the study's findings may not be applicable to diverse areas and regions.”
Comment 5. I still cannot find the lipid targets for your inclusion criteria, or any definition for this. I understand, that you used the guideline mentioned goals for the diagnosis and treatment, but you mentioned in your answer, that there were patients with long-standing and patients with new-onset of hyperlipidemias. At least, this fact should be mentioned in the manuscript, whether you cannot afford a value.
Answer 5. Thank you for the amendment. The manuscript was completed. (Line 81)
„There were patients with both long-standing and new-onset hyperlipidemias.”
Results:
Comment 6. In line 105 is still Pearson correlation. Please correct it!
Answer 6. Again, we agree with your observation, and correction was made accordingly (line 106).
Comment 7. The Spearman’s correlation is a better tool for analyze your data, but still the presentation in the text and the table you have inserted is not about the same. Your Table 5 shows data about the association between age and the parameters (congestive heart failure, visceral fat, physical activity), and not about congestive heart failure and visceral fat. Anyway, the analyzes about visceral fat and age, Pearson was a good choice, as these parameters are measured at the continuous scale.
Answer 7. Thank you very much for the comment. I corrected the manuscript. (lines 264, 268)
Comment 8. Thank you for rewrite the chi square test results, but it still not suitable. You now describe in the text and also in the table the same information, which is not necessary. Please provide in the Table 2 and 4 the number (N) of patients in each groups, or at least the percentage. Like you did it in case of Table 3.
Answer 8. Thank you for the amendment. We, the authors, have corrected them.
Comment 9. You have a binary scale for describing physical activity. In this case I don’t understand the phrase “mean” physical activity. I think you wanted to describe the average number of patients who are participating in physical activity and not the average of a nominal variable. Please correct the graph, maybe it would be better to add some units to avoid misunderstandings.
Answer 8. Thank you for the amendment. We, the authors, have corrected them.
Comment 9. For the Cox regression: Thank you for taking time to perform this analyzes. What is the time-to event parameter in your test? If it is new onset of congestive heart failure, it could not be also a variable in the equation. Please revise the description and your results!
Answer 9. Thank you for the suggestion. We, the authors, have completed the manuscript with the request information. (lines 283-287)
In the Cox regression calculation, a 1.7% decrease in age signaled a 6% increase in the risk rate of congestive heart failure (ExpB: 1.77, 95% CI: 1.27–2.47 and P= 0.001) independent of a number of potential confounders, including age, sex, HTN, and dyslipidemias. The time to event parameter in our test is one year.
Sincerely Yours,
